# Adversarial Text Generation via Feature-Mover's Distance

**Liqun Chen**[1], **Shuyang Dai**[1], **Chenyang Tao**[1], **Dinghan Shen**[1],
**Zhe Gan**[2], **Haichao Zhang**[4], **Yizhe Zhang**[3], **Lawrence Carin**[1]
[1]Duke University, [2]Microsoft Dynamics 365 AI Research, [3]Microsoft Research, [4]Baidu Research
`liqun.chen@duke.edu`

## Abstract

Generative adversarial networks (GANs) have achieved significant success in generating real-valued data. However, the discrete nature of text hinders the application of GAN to text-generation tasks. Instead of using the standard GAN objective, we propose to improve text-generation GAN via a novel approach inspired by optimal transport. Specifically, we consider matching the latent feature distributions of real and synthetic sentences using a novel metric, termed the feature-mover's distance (FMD). This formulation leads to a highly discriminative critic and easy-to-optimize objective, overcoming the mode-collapsing and brittle-training problems in existing methods. Extensive experiments are conducted on a variety of tasks to evaluate the proposed model empirically, including unconditional text generation, style transfer from non-parallel text, and unsupervised cipher cracking. The proposed model yields superior performance, demonstrating wide applicability and effectiveness.

## 1 Introduction

Natural language generation is an important building block in many applications, such as machine translation [5], dialogue generation [36], and image captioning [14]. While these applications demonstrate the practical value of generating coherent and meaningful sentences in a supervised setup, *unsupervised* text generation, which aims to estimate the distribution of real text from a corpus, is still challenging. Previous approaches, that often maximize the log-likelihood of each ground-truth word given prior observed words [41], typically suffer from exposure bias [6, 47], *i.e.*, the discrepancy between training and inference stages. During inference, each word is generated in sequence based on previously generated words, while during training ground-truth words are used for each timestep [27, 53, 58].

Recently, adversarial training has emerged as a powerful paradigm to address the aforementioned issues. The generative adversarial network (GAN) [21] matches the distribution of synthetic and real data by introducing a two-player adversarial game between a generator and a discriminator. The generator is trained to learn a nonlinear function that maps samples from a given (simple) prior distribution to synthetic data that appear realistic, while the discriminator aims to distinguish the fake data from real samples. GAN can be trained efficiently via back-propagation through the nonlinear function of the generator, which typically requires the data to be continuous (*e.g.*, images). However, the discrete nature of text renders the model non-differentiable, hindering use of GAN in natural language processing tasks.

Attempts have been made to overcome such difficulties, which can be roughly divided into two categories. The first includes models that combine ideas from GAN and reinforcement learning (RL), framing text generation as a sequential decision-making process. Specifically, the gradient of the generator is estimated via the policy-gradient algorithm. Prominent examples from this

category include SeqGAN [60], MaliGAN [8], RankGAN [37], LeakGAN [24] and MaskGAN [15]. Despite the promising performance of these approaches, one major disadvantage with such RL-based strategies is that they typically yield high-variance gradient estimates, known to be challenging for optimization [40, 61].

Models from the second category adopt the original framework of GAN without incorporating the RL methods (*i.e.*, RL-free). Distinct from RL-based approaches, TextGAN [61] and Gumbel-Softmax GAN (GSGAN) [31] apply a simple soft-argmax operator, and a similar Gumbel-softmax trick [28, 40], respectively, to provide a continuous approximation of the discrete distribution (*i.e.*, multinomial) on text, so that the model is still end-to-end differentiable. What makes this approach appealing is that it feeds the optimizer with low-variance gradients, improving stability and speed of training. In this work, we aim to improve the training of GAN that resides in this category.

When training GAN to generate text samples, one practical challenge is that the gradient from the discriminator often vanishes after being trained for only a few iterations. That is, the discriminator can easily distinguish the fake sentences from the real ones. TextGAN [61] proposed a remedy based on feature matching [49], adding Maximum Mean Discrepancy (MMD) to the original objective of GAN [22]. However, in practice, the model is still difficult to train. Specifically, (*i*) the bandwidth of the RBF kernel is difficult to choose; (*ii*) kernel methods often suffer from poor scaling; and (*iii*) empirically, TextGAN tends to generate short sentences.

In this work, we present feature mover GAN (FM-GAN), a novel adversarial approach that leverages optimal transport (OT) to construct a new model for text generation. Specifically, OT considers the problem of optimally transporting one set of data points to another, and is closely related to GAN. The earth-mover's distance (EMD) is employed often as a metric for the OT problem. In our setting, a variant of the EMD between the feature distributions of real and synthetic sentences is proposed as the new objective, denoted as the feature-mover's distance (FMD). In this adversarial game, the discriminator aims to maximize the dissimilarity of the feature distributions based on the FMD, while the generator is trained to minimize the FMD by synthesizing more-realistic text. In practice, the FMD is turned into a differentiable quantity and can be computed using the proximal point method [59].

The main contributions of this paper are as follows: (*i*) A new GAN model based on optimal transport is proposed for text generation. The proposed model is RL-free, and uses a so-called feature-mover's distance as the objective. (*ii*) We evaluate our model comprehensively on unconditional text generation. When compared with previous methods, our model shows a substantial improvement in terms of generation quality based on the BLEU statistics [43] and human evaluation. Further, our model also achieves good generation diversity based on the self-BLEU statistics [63]. (*iii*) In order to demonstrate the versatility of the proposed method, we also generalize our model to conditional-generation tasks, including non-parallel text style transfer [54], and unsupervised cipher cracking [20].

## 2 Background

### 2.1 Adversarial training for distribution matching

We review the basic idea of adversarial distribution matching (ADM), which avoids the specification of a likelihood function. Instead, this strategy defines draws from the synthetic data distribution $p_G(\boldsymbol{x})$ by drawing a latent code $\boldsymbol{z} \sim p(\boldsymbol{z})$ from an easily sampled distribution $p(\boldsymbol{z})$, and learning a generator function $G(\boldsymbol{z})$ such that $\boldsymbol{x} = G(\boldsymbol{z})$. The form of $p_G(\boldsymbol{x})$ is neither specified nor learned, rather we learn to draw samples from $p_G(\boldsymbol{x})$. To match the ensemble of draws from $p_G(\boldsymbol{x})$ with an ensemble of draws from the real data distribution $p_d(\boldsymbol{x})$, ADM introduces a variational function $\mathbb{V}(p_d, p_G; D)$, where $D(\boldsymbol{x})$ is known as the critic function or discriminator. The goal of ADM is to obtain an equilibrium of the following objective:

$$\min_G \max_D \mathbb{V}(p_d, p_G; D),\qquad(1)$$

where $\mathbb{V}(p_d, p_G; D)$ is computed using samples from $p_d$ and $p_G$ (*not* explicitly in terms of the distributions themselves), and $d(p_d, p_G) = \max_D \mathbb{V}(p_d, p_G; D)$ defines a discrepancy metric between two distributions [3, 42]. One popular example of ADM is the generative adversarial network (GAN), in which $\mathbb{V}_{\text{JSD}} = \mathbb{E}_{\boldsymbol{x} \sim p_d(\boldsymbol{x})} \log D(\boldsymbol{x}) + \mathbb{E}_{\boldsymbol{z} \sim p(\boldsymbol{z})} \log[1 - D(G(\boldsymbol{z}))]$ recovers the Jensen-Shannon divergence (JSD) for $d(p_d, p_G)$ [21]; expectations $\mathbb{E}_{\boldsymbol{x} \sim p_d(\boldsymbol{x})}(\cdot)$ and $\mathbb{E}_{\boldsymbol{z} \sim p(\boldsymbol{z})}(\cdot)$ are computed approximately with samples from the respective distributions. Most of the existing work in applying GAN

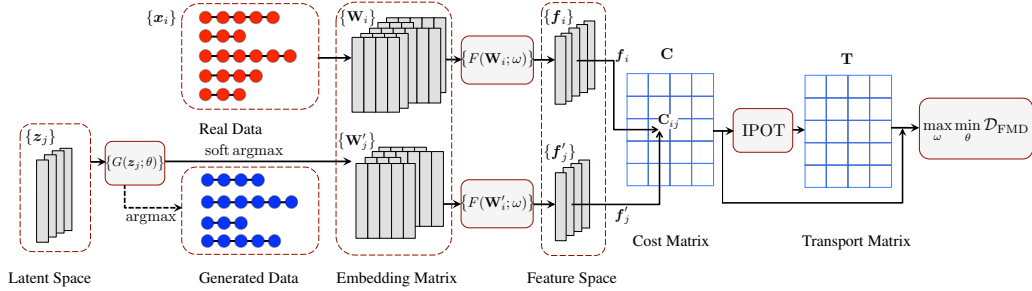

Figure 1: Illustration of the proposed feature mover GAN (FM-GAN) for text generation.

for text generation also uses this standard form, by combining it with policy gradient [60]. However, it has been shown in [2] that this standard GAN objective suffers from an unstably weak learning signal when the discriminator gets close to local optimal, due to the gradient-vanishing effect. This is because the JSD implied by the original GAN loss is not continuous wrt the generator parameters.

## 2.2 Sentence to feature

GAN models were originally developed for learning to draw from a continuous distribution. The discrete nature of text samples hinders the use of GANs, and thus a vectorization of a sequence of discrete tokens is considered. Let $\boldsymbol{x} = \{\boldsymbol{s}_1, ..., \boldsymbol{s}_L\} \in \mathbb{R}^{v \times L}$ be a sentence of length $L$, where $\boldsymbol{s}_t \in \mathbb{R}^v$ denotes the one-hot representation for the $t$-th word. A word-level vector representation of each word in $\boldsymbol{x}$ is achieved by learning a word embedding matrix $\mathbf{W}_{\boldsymbol{e}} \in \mathbb{R}^{k \times v}$, where $v$ is the size of the vocabulary. Each word is represented as a $k$-dimensional vector $\boldsymbol{w}_t = \mathbf{W}_{\boldsymbol{e}} \boldsymbol{s}_t \in \mathbb{R}^k$. The sentence $\boldsymbol{x}$ is now represented as a matrix $\mathbf{W} = [\boldsymbol{w}_1, ..., \boldsymbol{w}_L] \in \mathbb{R}^{k \times L}$. A neural network $F(\cdot)$, such as RNN [5, 10], CNN [18, 29, 52] or SWEM [51], can then be applied to extract feature vector $\boldsymbol{f} = F(\mathbf{W})$.

## 2.3 Optimal transport

GAN can be interpreted in the framework of optimal transport theory, and it has been shown that the *Earth-Mover's Distance* (EMD) is a good objective for generative modeling [3]. Originally applied in content-based image retrieval tasks [48], EMD is well-known for comparing multidimensional distributions that are used to describe the different features of images (*e.g.*, brightness, color, and texture content). It is defined as the ground distance (*i.e.*, cost function) between every two perceptual features, extending the notion of a distance between single elements to a distance between sets of elements. Specifically, consider two probability distribution $x \sim \mu$ and $y \sim \nu$; EMD can be then defined as:

$$\mathcal{D}_{\text{EMD}}(\mu, \nu) = \inf_{\gamma \in \Pi(\mu, \nu)} \mathbb{E}_{(x,y) \sim \gamma} \ c(x, y), \qquad (2)$$

where $\Pi(\mu, \nu)$ denotes the set of all joint distributions $\gamma(x, y)$ with marginals $\mu(x)$ and $\nu(y)$, and $c(x, y)$ is the cost function (*e.g.*, Euclidean or cosine distance). Intuitively, EMD is the minimum cost that $\gamma$ has to transport from $\mu$ to $\nu$.

## 3 Feature Mover GAN

We propose a new GAN framework for discrete text data, called feature mover GAN (FM-GAN). The idea of optimal transport (OT) is integrated into adversarial distribution matching. Explicitly, the original critic function in GANs is replaced by the Earth-Mover's Distance (EMD) between the sentence features of real and synthetic data. In addition, to handle the intractable issue when computing (2) [3, 49], we define the Feature-Mover's Distance (FMD), a variant of EMD that can be solved tractably using the Inexact Proximal point method for OT (IPOT) algorithm [59]. In the following sections, we discuss the main objective of our model, the detailed training process for text generation, as well as extensions. Illustration of the framework is shown in Figure 1.

### 3.1 Feature-mover's distance

In practice, it is not tractable to calculate the minimization over $\gamma$ in (2) [3, 19, 50]. In this section, we propose the *Feature-Mover's Distance* (FMD) which can be solved tractably. Consider two sets of sentence feature vectors $\mathbf{F} = \{\boldsymbol{f}_i\}_{i=1}^m \in \mathbb{R}^{d \times m}$ and $\mathbf{F}' = \{\boldsymbol{f}'_j\}_{j=1}^n \in \mathbb{R}^{d \times n}$ drawn from two different sentence feature distributions $\mathbb{P}_{\boldsymbol{f}}$ and $\mathbb{P}_{\boldsymbol{f}'}$; $m$ and $n$ are the total number of $d$-dimensional sentence features in $\mathbf{F}$ and $\mathbf{F}'$, respectively. Let $\mathbf{T} \in \mathbb{R}^{m \times n}$ be a transport matrix in which $\mathbf{T}_{ij} \geq 0$ defines how much of feature vector $\boldsymbol{f}_i$ would be transformed to $\boldsymbol{f}'_j$. The FMD between two sets of sentence features is then defined as:

$$\mathcal{D}_{\text{FMD}}(\mathbb{P}_{\boldsymbol{f}}, \mathbb{P}_{\boldsymbol{f}'}) = \min_{\mathbf{T} \geq 0} \sum_{i=1}^m \sum_{j=1}^n \mathbf{T}_{ij} \cdot c(\boldsymbol{f}_i, \boldsymbol{f}'_j) = \min_{\mathbf{T} \geq 0} \langle \mathbf{T}, \mathbf{C} \rangle, \qquad (3)$$

where $\sum_{j=1}^n \mathbf{T}_{ij} = \frac{1}{m}$ and $\sum_{i=1}^m \mathbf{T}_{ij} = \frac{1}{n}$ are the constraints, and $\langle \cdot, \cdot \rangle$ represents the Frobenius dot-product. In this work, the transport cost is defined as the cosine distance: $c(\boldsymbol{f}_i, \boldsymbol{f}'_j) = 1 - \frac{\boldsymbol{f}_i^\top \boldsymbol{f}'_j}{\|\boldsymbol{f}_i\|_2 \|\boldsymbol{f}'_j\|_2}$, and $\mathbf{C}$ is the cost matrix such that $\mathbf{C}_{ij} = c(\boldsymbol{f}_i, \boldsymbol{f}'_j)$. Note that during training, we set $m = n$ as the mini-batch size.

We propose to use the Inexact Proximal point method for Optimal Transport (IPOT) algorithm to compute the optimal transport matrix $\mathbf{T}^*$, which provides a solution to the original optimal transport problem (3) [59]. Specifically, IPOT iteratively solves the following optimization problem:

$$\mathbf{T}^{(t+1)} = \text{argmin}_{\mathbf{T} \in \Pi(\boldsymbol{f}, \boldsymbol{f}')} \langle \mathbf{T}, \mathbf{C} \rangle + \beta \mathbb{D}_h(\mathbf{T}, \mathbf{T}^{(t)}), \qquad (4)$$

where $\mathbb{D}_h(\mathbf{T}, \mathbf{T}^{(t)}) = \sum_{i,j} \mathbf{T}_{ij} \log \frac{\mathbf{T}_{ij}}{\mathbf{T}_{ij}^{(t)}} - \sum_{i,j} \mathbf{T}_{ij} + \sum_{i,j} \mathbf{T}_{ij}^{(t)}$ denotes the Bregman divergence wrt the entropy functional $h(\mathbf{T}) = \sum_{i,j} \mathbf{T}_{ij} \log \mathbf{T}_{ij}$.

Here the Bregman divergence $\mathbb{D}_h$ serves as a proximity metric and $\beta$ is the proximity penalty. This problem can be solved efficiently by Sinkhorn-style proximal point iterations [13, 59], as detailed in Algorithm 1.

Notably, unlike the Sinkhorn algorithm [19], we do not need to back-propagate the gradient through the proximal point iterations, which is justified by the Envelope Theorem [1] (see the Supplementary Material (SM)). This accelerates the learning process significantly and improves training stability [59].

---

**Algorithm 1** IPOT algorithm [59]

1: **Input:** batch size $n$, $\{\boldsymbol{f}_i\}_{i=1}^n, \{\boldsymbol{f}'_j\}_{j=1}^n, \beta$
2: $\boldsymbol{\sigma} = \frac{1}{n}\mathbf{1_n}, \mathbf{T}^{(1)} = \mathbf{11}^\top$
3: $\mathbf{C}_{ij} = c(\boldsymbol{f}_i, \boldsymbol{f}'_j), \mathbf{A}_{ij} = \text{e}^{-\frac{\mathbf{C}_{ij}}{\beta}}$
4: **for** $t = 1, 2, 3 \ldots$ **do**
5: $\quad \mathbf{Q} = \mathbf{A} \odot \mathbf{T}^{(t)}$ // $\odot$ is Hadamard product
6: $\quad$ **for** $k = 1, 2, 3, \ldots K$ **do**
7: $\qquad \boldsymbol{\delta} = \frac{1}{n\mathbf{Q}\boldsymbol{\sigma}}, \boldsymbol{\sigma} = \frac{1}{n\mathbf{Q}^\top \boldsymbol{\delta}}$
8: $\quad$ **end for**
9: $\quad \mathbf{T}^{(t+1)} = \text{diag}(\boldsymbol{\delta})\mathbf{Q}\text{diag}(\boldsymbol{\sigma})$
10: **end for**

---

**Algorithm 2** Adversarial text generation via FMD.

1: **Input:** batch size $n$, dataset $\mathbf{X}$, learning rate $\eta$, maximum number of iterations $N$.
2: **for** itr $= 1, \ldots N$ **do**
3: $\quad$ **for** $j = 1, \ldots, J$ **do**
4: $\qquad$ Sample a mini-batch of $\{\boldsymbol{x}_i\}_1^n \sim \mathbf{X}$ and $\{\boldsymbol{z}_i\}_1^n \sim \mathcal{N}(\mathbf{0}, \mathbf{I})$;
5: $\qquad$ Extract sentence features $\mathbf{F} = \{F(\mathbf{W}_e \boldsymbol{x}_i; \phi)\}_1^n$ and $\mathbf{F}' = \{F(G(\boldsymbol{z}_i; \theta); \phi)\}_1^n$;
6: $\qquad$ Update the feature extractor $F(\cdot; \phi)$ by maximizing:

$$\mathcal{L}_{\text{FM-GAN}}(\{\boldsymbol{x}_i\}_1^n, \{\boldsymbol{z}_i\}_1^n; \phi) = \mathcal{D}_{\text{FMD}}(\mathbf{F}, \mathbf{F}'; \phi)$$

7: $\quad$ **end for**
8: $\quad$ Repeat Step 4 and 5;
9: $\quad$ Update the generator $G(\cdot; \theta)$ by minimizing:

$$\mathcal{L}_{\text{FM-GAN}}(\{\boldsymbol{x}_i\}_1^n, \{\boldsymbol{z}_i\}_1^n; \theta) = \mathcal{D}_{\text{FMD}}(\mathbf{F}, \mathbf{F}'; \theta)$$

10: **end for**

---

## 3.2 Adversarial distribution matching with FMD

To integrate FMD into adversarial distribution matching, we propose to solve the following mini-max game:

$$\min_G \max_F \mathcal{L}_{\text{FM-GAN}} = \min_G \max_F \mathbb{E}_{\boldsymbol{x} \sim p_{\boldsymbol{x}}, \boldsymbol{z} \sim p_{\boldsymbol{z}}} \left[ \mathcal{D}_{\text{FMD}}(F(\mathbf{W}_{\boldsymbol{e}} \boldsymbol{x}), F(G(\boldsymbol{z}))) \right], \tag{5}$$

where $F(\cdot)$ is the sentence feature extractor, and $G(\cdot)$ is the sentence generator. We call this feature mover GAN (FM-GAN). The detailed training procedure is provided in Algorithm 2.

**Sentence generator** The Long Short-Term Memory (LSTM) recurrent neural network [25] is used as our sentence generator $G(\cdot)$ parameterized by $\theta$. Let $\mathbf{W}_{\boldsymbol{e}} \in \mathbb{R}^{k \times v}$ be our learned word embedding matrix, where $v$ is the vocabulary size, with each word in sentence $\boldsymbol{x}$ embedded into $\boldsymbol{w}_t$, a $k$-dimensional word vector. All words in the synthetic sentence are generated sequentially, *i.e.*,

$$\boldsymbol{w}_t = \mathbf{W}_{\boldsymbol{e}} \operatorname{argmax}(\boldsymbol{a}_t), \quad \text{where} \quad \boldsymbol{a}_t = \mathbf{V} \boldsymbol{h}_t \in \mathbb{R}^v, \tag{6}$$

where $\boldsymbol{h}_t$ is the hidden unit updated recursively through the LSTM cell: $\boldsymbol{h}_t = \text{LSTM}(\boldsymbol{w}_{t-1}, \boldsymbol{h}_{t-1}, \boldsymbol{z})$, $\mathbf{V}$ is a decoding matrix, softmax$(\mathbf{V}\boldsymbol{h}_t)$ defines the distribution over the vocabulary. Note that, distinct from a traditional sentence generator, here, the *argmax* operation is used, rather than sampling from a multinomial distribution, as in the standard LSTM. Therefore, all randomness during the generation is clamped into the noise vector $\boldsymbol{z}$.

The generator $G$ cannot be trained, due to the non-differentiable function argmax. Instead, an *soft-argmax* operator [61] is used as a continuous approximation:

$$\tilde{\boldsymbol{w}}_t = \mathbf{W}_{\boldsymbol{e}} \operatorname{softmax}(\tilde{\boldsymbol{a}}_t), \quad \text{where} \quad \tilde{\boldsymbol{a}}_t = \mathbf{V} \boldsymbol{h}_t / \tau \in \mathbb{R}^v, \tag{7}$$

where $\tau$ is the temperature parameter. Note when $\tau \to 0$, this approximates (6). We denote $G(\boldsymbol{z}) = (\tilde{\boldsymbol{w}}_1, \ldots, \tilde{\boldsymbol{w}}_L) \in \mathbb{R}^{k \times L}$ as the approximated embedding matrix for the synthetic sentence.

**Feature extractor** We use the convolutional neural network proposed in [11, 29] as our sentence feature extractor $F(\cdot)$ parameterized by $\phi$, which contains a convolution layer and a max-pooling layer. Assuming a sentence of length $L$, the sentence is represented as a matrix $\mathbf{W} \in \mathbb{R}^{k \times L}$, where $k$ is the word-embedding dimension, and $L$ is the maximum sentence length. A convolution filter $\mathbf{W}_{conv} \in \mathbb{R}^{k \times l}$ is applied to a window of $l$ words to produce new features. After applying the nonlinear activation function, we then use the max-over-time pooling operation [11] to the feature maps and extract the maximum values. While the convolution operator can extract features independent of their positions in the sentence, the max-pooling operator tries to capture the most salient features.

The above procedure describes how to extract features using one filter. Our model uses multiple filters with different window sizes, where each filter is considered as a linguistic feature detector. Assume $d_1$ different window sizes, and for each window size we have $d_2$ filters; then a sentence feature vector can be represent as $\boldsymbol{f} = F(\mathbf{W}) \in \mathbb{R}^d$, where $d = d_1 \times d_2$.

## 3.3 Extensions to conditional text generation tasks

**Style transfer** Our FM-GAN model can be readily generalized to conditional generation tasks, such as text style transfer [26, 35, 44, 54]. The style transfer task is essentially learning the conditional distribution $p(\boldsymbol{x}_2 | \boldsymbol{x}_1; c_1, c_2)$ and $p(\boldsymbol{x}_1 | \boldsymbol{x}_2; c_1, c_2)$, where $c_1$ and $c_2$ represent the labels for different styles, with $\boldsymbol{x}_1$ and $\boldsymbol{x}_2$ sentences in different styles. Assuming $\boldsymbol{x}_1$ and $\boldsymbol{x}_2$ are conditionally independent given the latent code $\boldsymbol{z}$, we have:

$$p(\boldsymbol{x}_1 | \boldsymbol{x}_2; c_1, c_2) = \int_{\boldsymbol{z}} p(\boldsymbol{x}_1 | \boldsymbol{z}, c_1) \cdot p(\boldsymbol{z} | \boldsymbol{x}_2, c_2) d\boldsymbol{z} = \mathbb{E}_{\boldsymbol{z} \sim p(\boldsymbol{z} | \boldsymbol{x}_2, c_2)} [p(\boldsymbol{x}_1 | \boldsymbol{z}, c_1)]. \tag{8}$$

Equation (8) suggests an autoencoder can be applied for this task. From this perspective, we can apply our optimal transport method in the cross-aligned autoencoder [54], by replacing the standard GAN loss with our FMD critic. We follow the same idea as [54] to build the style transfer framework. $E : \mathcal{X} \times \mathcal{C} \to \mathcal{Z}$ is our encoder that infers the content $\boldsymbol{z}$ from given style $c$ and sentence $\boldsymbol{x}$; $G : \mathcal{Z} \times \mathcal{C} \to \mathcal{X}$ is our decoder that generates synthetic sentence $\hat{\boldsymbol{x}}$, given content $\boldsymbol{z}$ and style $c$. We add the following reconstruction loss for the autoencoder:

$$\mathcal{L}_{\text{rec}} = \mathbb{E}_{\boldsymbol{x}_1 \sim p_{\boldsymbol{x}_1}} \left[ -\log p_G(\boldsymbol{x}_1 | c_1, E(\boldsymbol{x}_1, c_1)) \right] + \mathbb{E}_{\boldsymbol{x}_2 \sim p_{\boldsymbol{x}_2}} \left[ -\log p_G(\boldsymbol{x}_2 | c_2, E(\boldsymbol{x}_2, c_2)) \right], \tag{9}$$

where $p_{\boldsymbol{x}_1}$ and $p_{\boldsymbol{x}_2}$ are the empirical data distribution for each style. We also need to implement adversarial training on the generator $G$ with discrete data. First, we use the soft-argmax approximation discussed in Section 3.2; second, we also use Professor-Forcing [32] algorithm to match the sequence of LSTM hidden states. That is, the discriminator is designed to discriminate $\hat{\boldsymbol{x}}_2 = G(E(\boldsymbol{x}_1, c_1), c_2)$ with real sentence $\boldsymbol{x}_2$. Unlike [54] which uses two discriminators, our model only needs to apply the FMD critic twice to match the distributions for two different styles:

$$\mathcal{L}_{\text{adv}} = \mathbb{E}_{\boldsymbol{x}_1 \sim p_{\boldsymbol{x}_1}, \boldsymbol{x}_2 \sim p_{\boldsymbol{x}_2}} [\mathcal{D}_{\text{FMD}}(F(G(E(\boldsymbol{x}_1, c_1), c_2)), F(\mathbf{W}_e \boldsymbol{x}_2)) \quad (10)$$
$$+ \mathcal{D}_{\text{FMD}}(F(G(E(\boldsymbol{x}_2, c_2), c_1)), F(\mathbf{W}_e \boldsymbol{x}_1))] ,$$

where $\mathbf{W}_e$ is the learned word embedding matrix. The final objective function for this task is: $\min_{G,E} \max_F \mathcal{L}_{\text{rec}} + \lambda \cdot \mathcal{L}_{\text{adv}}$, where $\lambda$ is a hyperparameter that balances these two terms.

**Unsupervised decipher**    Our model can also be used to tackle the task of unsupervised cipher cracking by using the framework of CycleGAN [62]. In this task, we have two different corpora, *i.e.*, $\mathbf{X}_1$ denotes the original sentences, and $\mathbf{X}_2$ denotes the encrypted corpus using some cipher code, which is unknown to our model. Our goal is to design two generators that can map one corpus to the other, *i.e.*, $G_1 : \mathbf{X}_1 \to \mathbf{X}_2$, $G_2 : \mathbf{X}_2 \to \mathbf{X}_1$. Unlike the style-transfer task, we define $F_1$ and $F_2$ as two sentence feature extractors for the different corpora. Here we denote $p_{\boldsymbol{x}_1}$ to be the empirical distribution of the original corpus, and $p_{\boldsymbol{x}_2}$ to be the distribution of the encrypted corpus. Following [20], we design two losses: the cycle-consistency loss (reconstruction loss) and the adversarial feature matching loss. The cycle-consistency loss is defined on the feature space as:

$$\mathcal{L}_{\text{cyc}} = \mathbb{E}_{\boldsymbol{x}_1 \sim p_{\boldsymbol{x}_1}} [\|F_1(G_2(G_1(\boldsymbol{x}_1))) - F_1(\mathbf{W}_e \boldsymbol{x}_1)\|_1] + \mathbb{E}_{\boldsymbol{x}_2 \sim p_{\boldsymbol{x}_2}} [\|F_2(G_1(G_2(\boldsymbol{x}_2))) - F_2(\mathbf{W}_e \boldsymbol{x}_2)\|_1] ,$$
$$(11)$$

where $\| \cdot \|_1$ denotes the $\ell_1$-norm, and $\mathbf{W}_e$ is the word embedding matrix. The adversarial loss aims to help match the generated samples with the target:

$$\mathcal{L}_{\text{adv}} = \mathbb{E}_{\boldsymbol{x}_1 \sim p_{\boldsymbol{x}_1}, \boldsymbol{x}_2 \sim p_{\boldsymbol{x}_2}} [\mathcal{D}_{\text{FMD}}(F_1(G_2(\boldsymbol{x}_2)), F_1(\mathbf{W}_e \boldsymbol{x}_1)) + \mathcal{D}_{\text{FMD}}(F_2(G_1(\boldsymbol{x}_1)), F_2(\mathbf{W}_e \boldsymbol{x}_2))] .$$
$$(12)$$

The final objective function for the decipher task is: $\min_{G_1,G_2} \max_{F_1,F_2} \mathcal{L}_{\text{cyc}} + \lambda \cdot \mathcal{L}_{\text{adv}}$, where $\lambda$ is a hyperparameter that balances the two terms.

## 4    Related work

**GAN for text generation**    SeqGAN [60], MaliGAN [8], RankGAN [37], and MaskGAN [15] use reinforcement learning (RL) algorithms for text generation. The idea behind all these works are similar: they use the REINFORCE algorithm to get an unbiased gradient estimator for the generator, and apply the roll-out policy to obtain the reward from the discriminator. LeakGAN [24] adopts a hierarchical RL framework to improve text generation. However, it is slow to train due to its complex design. For GANs in the RL-free category, GSGAN [31] and TextGAN [61] use the Gumbel-softmax and soft-argmax trick, respectively, to deal with discrete data. While the latter uses MMD to match the features of real and synthetic sentences, both models still keep the original GAN loss function, which may result in the gradient-vanishing issue of the discriminator.

**GAN with OT**    Wasserstein GAN (WGAN) [3, 23] applies the EMD by imposing the 1-*Lipschitz* constraint on the discriminator, which alleviates the gradient-vanishing issue when dealing with continuous data (*i.e.*, images). However, for discrete data (*i.e.*, text), the gradient still vanishes after a few iterations, even when weight-clipping or the gradient-penalty is applied on the discriminator [20]. Instead, the Sinkhorn divergence generative model (Sinkhorn-GM) [19] and Optimal transport GAN (OT-GAN) [50] optimize the Sinkhorn divergence [13], defined as an entropy regularized EMD (2): $W_\epsilon(\boldsymbol{f}, \boldsymbol{f}') = \min_{T \in \Pi(\boldsymbol{f}, \boldsymbol{f}')} \langle \mathbf{T}, \mathbf{C} \rangle + \epsilon \cdot h(\mathbf{T})$, where $h(\mathbf{T}) = \sum_{i,j} \mathbf{T}_{ij} \log \mathbf{T}_{ij}$ is the entropy term, and $\epsilon$ is the hyperparameter. While the Sinkhorn algorithm [13] is proposed to solve this entropy regularized EMD, the solution is sensitive to the value of the hyperparameter $\epsilon$, leading to a trade-off between computational efficiency and training stability. Distinct from that, our method uses IPOT to tackle the original problem of OT. In practice, IPOT is more efficient than the Sinkhorn algorithm, and the hyperparameter $\beta$ in (4) only affects the convergence rate [59].

## 5    Experiment

We apply the proposed model to three application scenarios: generic (unconditional) sentence generation, conditional sentence generation (with pre-specified sentiment), and unsupervised decipher.

| Dataset | Train | Test | Vocabulary | average length |
|---|---|---|---|---|
| CUB captions | 100,000 | 10,000 | 4,391 | 15 |
| MS COCO captions | 120,000 | 10,000 | 27,842 | 11 |
| EMNLP2017 WMT News | 278,686 | 10,000 | 5,728 | 28 |

Table 1: Summary statistics for the datasets used in the generic text generation experiments.

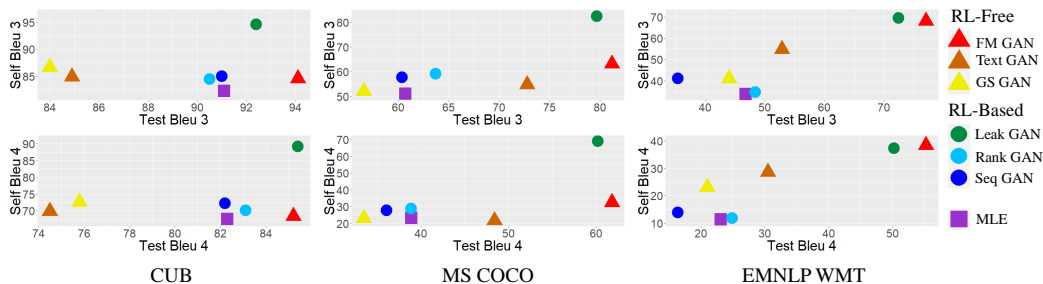

CUB      MS COCO      EMNLP WMT

Figure 2: Test-BLEU score (higher value implies better quality) vs self-BLEU score (lower value implies better diversity). Upper panel is BLEU-3 and lower panel is BLEU-4.

For the generic sentence generation task, we experiment with three standard benchmarks: CUB captions [57], MS COCO captions [38], and EMNLP2017 WMT News [24].

Since the sentences in the CUB dataset are typically short and have similar structure, it is employed as our toy evaluation. For the second dataset, we sample $130,000$ sentences from the original MS COCO captions. Note that we do not remove any low-frequency words for the first two datasets, in order to evaluate the models in the case with a relatively large vocabulary size. The third dataset is a large long-text collection from EMNLP2017 WMT News Dataset. To facilitate comparison with baseline methods, we follow the same data preprocessing procedures as in [24]. The summary statistics of all the datasets are presented in Table 1.

For conditional text generation, we consider the task of transferring an original sentence to the opposite sentiment, in the case where parallel (paired) data are not available. We use the same data as introduced in [54]. For the unsupervised decipher task, we follow the experimental setup in CipherGAN [20] and evaluate the model improvement after replacing the critic with the proposed FMD objective.

We employ test-BLEU score [60], self-BLEU score [63], and human evaluation as the evaluation metrics for the generic sentence generation task. To ensure fair comparison, we perform extensive comparisons with several strong baseline models using the benchmark tool in Texygen [63]. For the non-parallel text style transfer experiment, following [26, 54], we use a pretrained classifier to calculate the sentiment accuracy of transferred sentences. We also leverage human evaluation to further measure the quality of the transferring results. For the deciphering experiment, we adopt the average proportion of correctly mapped words as accuracy as proposed in [20]. Our code will be released to encourage future research.

## 5.1 Generic text generation

In general, when evaluating the performance of different models, we desire high test-BLEU score (good quality) and low self-BLEU score (high diversity). Both scores should be considered: (*i*) a high test-BLEU score together with a high self-BLEU score means that the model might generate good sentences while suffering from mode collapse (*i.e.*, low diversity); (*ii*) if a model generates sentences randomly, the diversity of generated sentence could be high but the test-BLEU score would be low. Figure 2 is used to compare the performance of every model. For each subplot, the $x$-axis represents test-BLEU, and the $y$-axis represents self-BLEU (here we only show BLEU-3 and BLEU-4 figures; more quantitative results can be found in the SM). For the CUB and MS COCO datasets, our model achieves both high test-BLEU and low self-BLEU, providing realistic sentences with high diversity. For the EMNLP WMT dataset, the synthetic sentences from SeqGAN, RankGAN, GSGAN and

TextGAN is less coherent and realistic (examples can be found in the SM) due to the long-text nature of the dataset. In comparison, our model is still capable of providing realistic results.

To further evaluate the generation quality based on the EMNLP WMT dataset, we conduct a human Turing test on Amazon Mechanical Turk; 10 judges are asked to rate over 100 randomly sampled sentences from each

| Method | MLE | SeqGAN | RankGAN | LeakGAN |
|---|---|---|---|---|
| Human score | $2.54 \pm 0.79$ | $2.55 \pm 0.83$ | $2.86 \pm 0.95$ | $3.41 \pm 0.82$ |
| Method | GSGAN | TextGAN | Our model | real sentences |
| Human score | $2.52 \pm 0.78$ | $3.03 \pm 0.92$ | $\mathbf{3.72 \pm 0.80}$ | $4.21 \pm 0.77$ |

Table 2: Human evaluation results on EMNLP WMT.

model with a scale from 0 to 5. The means and standard deviations of the rating score are calculated and provided in Table 2. We also provide some examples of the generated sentences from LeakGAN and our model in Table 3. More generated sentences are provided in the SM.

| | |
|---|---|
| **LeakGAN**: | (1) " people , if aleppo recognised switzerland stability , " mr . trump has said that " " it has been filled before the courts . <br> (2) the russian military , meanwhile previously infected orders , but it has already been done on the lead of the attack . |
| **Ours**: | (1) this is why we will see the next few years , we ' re looking forward to the top of the world , which is how we ' re in the future . <br> (2) If you ' re talking about the information about the public , which is not available , they have to see a new study . |

Table 3: Examples of generated sentences from LeakGAN and our model.

## 5.2 Non-parallel text style transfer

Table 4 presents the sentiment transfer results on the Yelp review dataset, which is evaluated with the accuracy of transferred sentences, determined by a pretrained CNN classifier [29]. Note that with the same experimental setup as in [54], our model achieves significantly higher transferring accuracy compared with the cross-aligned autoencoder (CAE) model [54]. Moreover, our model even outperforms the controllable text generation method [26] and BST [44], where a sentiment classifier is directly pre-trained to guide the sentence generation process (on the contrary, our model is trained in an end-to-end manner and requires no pre-training steps), and thus should potentially have a better control over the style (*i.e.*, sentiment) of generated sentences [54]. The superior performance of the proposed method highlights the ability of FMD to mitigate the vanishing-gradient issue caused by the discrete nature of text samples, and give rises to better matching between the distributions of reviews belonging to two different sentiments.

Human evaluations are conducted to assess the quality of the transferred sentences. In this regard, we randomly sample 100 sentences from the test set, and 5 volunteers rate the outputs of different models in terms of their fluency, sentiment, and con-

| Method | Controllable [26] | CAE [54] | BST [44] | Our model |
|---|---|---|---|---|
| Accuracy(%) | 84.5 | 80.6 | 87.2 | **89.8** |
| Sentiment | 3.6 | 3.2 | - | **4.1** |
| Content | **4.6** | 4.1 | - | 4.5 |
| Fluency | 4.2 | 3.7 | - | **4.4** |

Table 4: Sentiment transfer accuracy and human evaluation results on Yelp.

tent preservation in a double blind fashion. The rating score is from 0 to 5. Detailed results are shown in Table 4. We also provide sentiment transfer examples in Table 5. More examples are provided in the SM.

| | |
|---|---|
| **Original**: | one of the best gourmet store shopping experiences i have ever had . |
| **Controllable** : | one of the best gourmet store shopping experiences i have ever had . |
| **CAE**: | one of the worst staff i would ever ever ever had ever had . |
| **Ours**: | one of the worst indian shopping store experiences i have ever had . |
| **Original**: | staff behind the deli counter were super nice and efficient ! |
| **Controllable**: | staff behind the deli counter were super rude and efficient ! |
| **CAE**: | the staff were the front desk and were extremely rude airport ! |
| **Ours**: | staff behind the deli counter were super nice and inefficient ! |

Table 5: Sentiment transfer examples.

### 5.3 Unsupervised decipher

CipherGAN [20] uses GANs to tackle the task of unsupervised cipher cracking, utilizing the framework of CycleGAN [62] and adopting techniques such as Gumbel-softmax [31] that deal with discrete data. The implication of unsupervised deciphering could be understood as unsupervised machine translation, in which one language might be treated as an enciphering of the other. In this experiment, we adapt the idea of feature mover's distance to the original framework of CipherGAN and test this modified model on the Brown English text dataset [16].

The Brown English-language corpus [30] has a vocabulary size of over one million. In this experiment, only the top 200 most frequent words are considered while the others are replaced by an "unknown" token. We denote this modified word-level dataset as Brown-W200. We use Vigenère [7] to encipher the original plain text. This dataset can be downloaded from this repository[1].

For fair comparison, all the model architectures and parameters are kept the same as CipherGAN while the critic for the discriminator is replaced by the FMD objective as shown in (3). Table 6 shows the quantitative results in terms of average proportion of words mapped in a given sequence (*i.e.*, deciphering accuracy). The baseline frequency analysis model only operates when the cipher key is known. Our model achieves higher accuracy compared to the original CipherGAN. Note that some other experimental setups from [20] are not evaluated, due to the extremely high accuracy (above 99%); the amount of improvement would not be apparent.

| Method | Freq. Analysis (with keys) | CipherGAN [20] | Our model |
|---|---|---|---|
| Accuracy(%) | < 0.1 (44.3) | 75.7 | 77.2 |

Table 6: Decipher results on Brown-W200.

## 6 Conclusion

We introduce a novel approach for text generation using feature-mover's distance (FMD), called feature mover GAN (FM-GAN). By applying our model to several tasks, we demonstrate that it delivers good performance compared to existing text generation approaches. For future work, FM-GAN has the potential to be applied on other tasks such as image captioning [56], joint distribution matching [9, 17, 34, 45, 46, 55], unsupervised sequence classification [39], and unsupervised machine translation [4, 12, 33].

## Acknowledgments

This research was supported in part by DARPA, DOE, NIH, ONR and NSF.

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
