[Reviews · NeurIPS 2018]

Reviewer 1



The authors introduce a new variation of GAN that is claimed to be suitable for text generation. The proposed method relies on a new optimal transport–based distance metric on the feature space learned by the “discriminator”. The idea is sound and seems to be novel. The text is well written and easy to follow. The authors present experimental results for unconditional text generation, style transfer from non-parallel text, and unsupervised cipher cracking. Overall, I like the ideas in the paper but I think that the experiments are not robust, which makes it difficult to judge if the current method represents a real advance over the previous GAN models for text generation. Some questions/comments about the experiments: (1) For the generic text generation, why not using datasets that have been used in other works: Penn Treebank, IMDB? (2) For generic text generation why the authors have not compared their results with MaskGAN? As far as I know, MaskGAN seems to be the state-of-the-art GAN for text generation. (3) For both generic text generation and style transfer, why the authors have not used the perplexity metric like in [40] and MaskGAN papers? Measuring the perplexity of generated text using a pretrained language model is an important complementary way to compare the models, specially the style transfer one. (4) The authors should be more specific regarding the setup of the human evaluations. The paper mentions “judges are asked to rate OVER 100 randomly sampled sentences”. Please tell the exact number of sentences. Each sentence was analyzed for how many judges? Moreover, the evaluation strategy itself seems far too subjective. If we look at the standard deviation, RankGAN, LeakGAN, TextGAN and the proposed method are on the same range of results. The correct evaluation should be done by showing the Turker a pair of sentences from two different models and asking him/her to select the sentence that he/she prefers. How does the training time of FMD tGAN compare to the other GAN models? --- The author feedback properly addressed my questions. Therefore, I’ve increased the overall score.

Reviewer 2



This paper proposes to improve the GAN model for text generation by introducing a novel metric called the feature-mover's distance (FMD). The metric is used to compare the latent feature distributions of the real and synthetic sentences. It is differentiable and can be computed via the proximal method. The FMD metric is a variant of the earth-mover's distance, often used in optimal transport. It measures the distance of feature distributions of real and synthetic sentences and is used as the new objective of the GAN model, where the discriminator aims to maximize the dissimilarity of the feature distributions and the generator is trained to do the opposite. The paper is well organized and clearly described in general. The experiments and evaluations are well thought-out. The proposed metric has been shown to perform well on three tasks, including both unconditional and conditional text generation tasks. Overall I found the paper to be an enjoyable read and think it would contribute nicely to the literature. I'd suggest the authors to perhaps also comment on the potential weaknesses of this approach. In particular, it looks like the generated sentences (in the Experiment section) appear to be less informative, and occasionally the system output has changed the meaning of the original sentence (in the style transfer case). I wonder why this is the case. The following paper may be relevant to the style transfer task: "Delete, Retrieve, Generate: a Simple Approach to Sentiment and Style Transfer", Robin Jia, He He, and Percy Liang, NAACL 2018.

Reviewer 3



The paper proposes a "feature-mover's distance" as the optimization objective of GAN applied to text. Experiments are conducted for text generation, text style transfer and text decipher tasks. The proposed idea is a variation on the choice of loss functions compared to previous GAN models for text. However, both the motivation and the experimental results are not significant enough to show an advantage of the proposed loss form. First of all, the paper mentions that "mode collapsing" and "brittle training" are 2 main problems facing GAN for text. These problems are true, but the paper shows no justification for why the proposed loss form could be helpful for mitigating these 2 problems, neither theoretically nor empirically. Secondly, since the main contribution is the loss form, the paper should have included fair comparisons with some other major loss forms using the same generator and discriminator network design. This is missing in the paper, and it is very hard to grasp why the particular choice of loss form is made. Besides, a recent paper [1] shows that loss choices of GAN do not matter too much for the generative results (albeit for image generation), which makes is even more necessary to include such comparisons in this paper. Thirdly, the evaluation results are not statistically significant. The first problem is the choice of evaluation metric (BLEU and self-BLEU), which are not well established scores for text generative models. It is questionable whether any form of BLEU is useful in this case, since it measures whether a phrase exists in a target sentence. Text generative models could simply generate phrases that could make sense, but beyond the measurement of such statistical matching scores because they do not exist as a target sample. One would argue that a large enough dataset would have solved this, but the datasets used in the paper are quite small compared to other available text corpora. Also, the self-BLEU score requires multiple generated texts, which is a hyper-parameter not clearly detailed in the paper. Furthermore, it is unclear whether the improvement obtained in the paper under such metric is significant enough to demonstrate an advantage of the proposed loss form. Finally, it is hard for me to perceive a difference from the generated samples in the supplemental material. The authors should perhaps consider including an explanation on why the proposed form generate better results from these samples. In a summary, due to the lack of proper motivation, comparison and significance in results, I recommend rejection for the paper in its current form. That said, the loss form is an interesting idea. If the authors can develop the idea further and show a clear advantage, it could perhaps be submitted again. [1] Mario Lucic, Karol Kurach, Marcin Michalski, Sylvain Gelly, Olivier Bousquet. Are GANs Created Equal? A Large-Scale Study. arXiv 1711.10337 ------ After reading the author rebuttal, I realized I missed figure 2 in the first round of reviewing. As a result, the main reason for recommending rejection -- lack of comparison with other GAN losses -- has been significantly weakened. I changed the score to recommend for acceptance now.